# The Expanded SWEET Gene Family Following Whole Genome Triplication in *Brassica rapa*

**DOI:** 10.3390/genes10090722

**Published:** 2019-09-18

**Authors:** Yanping Wei, Dong Xiao, Changwei Zhang, Xilin Hou

**Affiliations:** State Key Laboratory of Crop Genetics & Germplasm Enhancement/Key Laboratory of Biology and Genetic Improvement of Horticultural Crops (East China), Ministry of Agriculture and Rural Affairs of the P. R. China/Engineering Research Center of Germplasm Enhancement and Utilization of Horticultural Crops, Ministry of Education, Nanjing Agricultural University, Nanjing 210095, China; 2014204019@njau.edu.cn (Y.W.); dong.xiao@njau.edu.cn (D.X.); changweizh@njau.edu.cn (C.Z.)

**Keywords:** SWEETs, sugar transporter, *Brassica rapa*, evolutionary conservation, expression pattern

## Abstract

The SWEET family, which includes transcripts of a cohort of plant hexose and sucrose transporters, is considered key to improving crop stress tolerance and yield through its role in manipulating the carbohydrate partitioning process. The functions and regulatory roles of this gene family are variable among different species; thus, to determine these roles, more species-specific information is needed. *Brassica rapa* displays complicated regulation after a whole-genome triplication (WGT) event, which provides enormous advantages for use in genetic studies, thus it is an ideal model for exploring the functional and regulatory roles of SWEETs from a genetic perspective. In this study, the results of a homology search and phylogenetic relationship analysis revealed the evolutionary footprint of SWEETs among different plant taxa, which showed that plant SWEETs may have originated from Clade II and then expanded from vascular plants. The amino acid sequence characteristics and an analysis of the exon-intron structure of BrSWEETs duplicates clarified that SWEETs retention occurred after a WGT event in *B. rapa*. An analysis of the transcriptional levels of BrSWEETs in different tissues identified the expression differences among duplicated co-orthologs. In addition, qRT-PCR indicated that the BrSWEETs’ co-orthologs were varied in their stress responses. This study greatly enriches our knowledge of SWEETs in the *B. rapa* species, which will contribute to future studies on the *Brassica*-specific regulatory pathways and to creating genetic innovations.

## 1. Introduction 

Higher plants have evolved well for the ability to employ solar energy to convert CO_2_ into organic carbon in photosynthetic source leaf tissues, and this carbon then travels to various heterotrophic sink organs (such as seeds, roots, and fruits) [1]. The efficient distribution (translocation and partitioning) of sugars between a source–sink system is governed by sugar transporters, and it is crucial for sink organ development and positive feedback to source tissues to ensure sufficient energy allocation and to maintain a trade-off between the different parts of plants [1,2]. Therefore, sugar transporters usually act as bridges, and they are fundamentally pivotal to the cellular exchange of carbon and energy in multicellular organisms to perform further biological functions. Research has shown that membrane-localized sugar transporters in plants can be categorized into MFSs (the major facilitator superfamily) and SWEET (Sugars Will Eventually be Exported Transporter, PFAM: PF03083) family [3]. The homologs of the SWEET family among the higher plants are structurally characterized by seven transmembrane domains (TMDs) harboring two MtN3/PQ-loop domains connected by a loop, and they are phylogenetically divided into four clades with mono- and disaccharide transport activity by a uniport mechanism [4,5]. Their functions are essential for various activities in plants, such as phloem loading [6], pollen development [7], nectar secretion [8], seed filling [9,10,11], freezing tolerance [12], pathogen resistance [13,14,15], and others, and they act by mediating the translocation and partitioning of hexoses and/or sucrose.

SWEETs homologs have not only been found in prokaryotes (semiSWEETs) but have also reportedly appeared in all the eukaryotic kingdoms ranging from fungi to plants and animals [4,16,17]. Plant SWEETs were initially identified as sugar effluxers with glucose or sucrose transport activities in *Arabidopsis* and rice [5]. Thereafter, additional homologs of SWEETs from different plant varieties were explored and their functions were determined to improve crop survival and yield potential. For example, in the model plant *Arabidopsis*, plasma membrane (PM)-localized AtSWEET11 and AtSWEET12 of the clade III sucrose transporters were characterized by their role in the preparation of sucrose apoplastic phloem loading [18]. A double mutation (*Atsweet11/12*) blocks assimilation exudation, which results in a shorter root, abnormal vascular structure and lower freezing tolerance [6,18]. In addition, *AtSWEET11/12/15* are specifically expressed in developing seeds to transfer sugars from the seed coat to the embryo, and the triple mutant (*Atsweet11/12/15*) resulted in severe seed defects [19]. Like other members of clade III, AtSWEET9 has a demonstrated function as an essential sucrose efflux transporter in nectar production, and its orthologs are functionally conserved as nectary-specific transporters in core eudicots [8]; *AtSWEET10* is involved in the stress response, and *AtSWEET13* and *AtSWEET14* function within the anthers by tangling with the gibberellic acid response [20,21]. Apart from the sucrose transporters in Clade III, *AtSWEET1* [5] and *AtSWEET2* in clade I, *AtSWEET4* [22] and *AtSWEET8* [23,24] in clade II, and *AtSWEET16* [12] and *AtSWEET17* [25] in Clade IV, have been specifically explored for their significant functions in various biological activities through the manipulation of sugar transport. Parallel with the gradual growth of SWEETs research in *Arabidopsis*, the further study of SWEETs homologs in other plants, including rice [5,24,26,27], tomatoes [11,28,29], grapes [30,31], *Medicago truncatula* [32], etc., have been reported and rigorously updated. Therefore, given the versatile regulators and obvious yield potential of SWEETs, more detailed and in-depth studies in more abundant species are needed. 

The *Brassica rapa* plant provides a well-established model of the “U” triangle in the *Brassica* genus, which is characterized by broad genetic and morphological diversity that allowed for the domestication of this genus to produce leafy vegetables, vegetable oils, turnip roots, turnip greens, turnip tops, and fodder turnip [33,34,35]. The release of the *B. rapa* and *B. oleracea* genomes confirmed an additional γ whole-genome triplication (WGT) event in the *Brassica* species after they diverged from *Arabidopsis thaliana* approximately 9–28 million years ago [34,35,36,37,38,39]. In comparing the ~27,000 genes in the *A. thaliana* genome, the retained genes (~42,000) in the *B. rapa* genome are considerably fewer than the theoretical value from a simple WGT event. Approximately ~70%, ~46%, and 36% of genes remained in the sub-genomes of the least fractionated (LF), medium fractionated (MF1), and most fractionated (MF2) individuals, respectively, and they were distributed into 71 GBs (genetic blocks) comparing 24 conserved ancestral GBs in *A. thaliana* [34,40]. Previous reports have suggested that the reshuffle process, namely whole-genome duplication/triplication (WGD/WGT), subsequently intertwined the fractionation and re-diploidization that were ubiquitous in the plant kingdom, which had different impacts on the gene sets/family [41,42]. Structural and functional evidence has suggested that having multiple copies of SWEET genes in eukaryotes allows for diversification in metabolic regulation. However, the puzzle of the molecular characteristics of SWEETs in *B. rapa* remains unresolved.

With the aim of exploring the SWEET gene family in *B. rapa* (BrSWEETs), we investigated the evolutionary footprint of SWEETs in different plant taxa by homology search and phylogenetic relationship. Subsequently, the SWEETs were retained in *B. rapa* after the WGT event were analyzed using basic information, including its distribution on the chromosome, sub-genome, and 24 GB, and then it was characterized in detail by multiple sequence alignment, phylogenetic relationships, conserved domains, and amino acid motifs, along with exon-intron structure through comparisons with its homologs in *A. thaliana*. After that, the BrSWEETs’ transcriptional levels in different *B. rapa* tissues were investigated to illuminate the duplicated paralog expression. In addition, the WGD-duplicated SWEET’s paralogous expression under abiotic stress conditions was analyzed to understand the stress response pattern in *B. rapa*. Our fundamental work here will contribute to the growing body of knowledge centered on understanding the role of SWEET gene family expansion, and it may be valuable for target gene selection and further molecular function characterization to improve crop yields by modifying stress resistance.

## 2. Materials and Methods 

### 2.1. Identification and Characterization of SWEET Genes in B. rapa 

Homologous SWEETs in *B. rapa* were revealed in the BRAD database [43] according to a syntenic matchup with SWEET isoforms in *A. thaliana*, and they were subsequently confirmed by local BLASTP alignment (*E*-value ≤ 1e−5 and identity ≥ 60%) [44] and PFAM domain analysis (https://pfam.xfam.org). The position of each identified member on the ancestral karyotypes, genetic blocks, and sub-genomes (LF, MF1, and MF2), along with the exon-intron structure was verified by searching the BRAD database (*B. rapa* genome v1.5, accession Chinese cabbage Chiifu-401-42, http://brassicadb.org/brad/index.php). The MCScanX (Multiple Collinearity Scan toolkit) (http://chibba.pgml.uga.edu/mcscan2/; parameter settings of match_score: 50, match_size: 5, gap_score: −3, and *E*-value: 1e−5) was employed to identify duplication events for the SWEETs families in *A. thaliana* and *B. rapa.* A TMHMM server (V.2.0) (http://www.cbs.dtu.dk/services/TMHMM/) was used to determine the transmembrane helices (TMs). The conserved domain and motif analysis were subsequently performed using online CDD tools (Conserved Domains Database) (https://www.ncbi.nlm.nih.gov/Structure/bwrpsb/bwrpsb.cgi) and the MEME program (http://meme-suite.org/tools/meme).

### 2.2. Multiple Sequence Alignment and Phylogenetic Analysis

For the homologous SWEETs’ analyses in *B. rapa* and *A. thaliana*, a multiple sequence alignment of the amino acid sequence was performed with the MAFFT program (v7.245) [45] and was demonstrated by TEXshade (www.pharmazie.uni-kiel.de/chem/Prof Beitz/biotex.html) [46]. The seven TM domains were assigned using a sequence alignment based on the *AtSWEET1* structure indicated by Tao et al. [4]. The phylogenetic relationship was inferred using the NJ (neighbor-joining) method [47] based on the optimal model (JTT+G) with the support of 1000 bootstrap replications with a MEGA program [48]. 

Additionally, to investigate the footprint of SWEET genes in different plant groups, the homologs were retrieved from the PLAZA database by BLASTP search strategy (parameter cutoff setting: *E*-value ≤ 1e−10 and identity ≥ 60%) [44]. Only the candidate homologs with two MtN3/PQ-loop domains harboring seven TMs structure were further used to perform the phylogenetic analysis. The ML (maximum likelihood) phylogenetic trees were constructed using PhyML-3.1 software [49], after multiple sequence alignments with the MAFFT program (v7.245) [45] by setting up the parameters as follows: 1000 bootstrap replications; NJ/BioNJ as the initial tree; the optimal model (JTT+G) determined by the Protest program (v3.2) [50] according to BIC (Bayesian Information Criterion); and everything else according to default settings [51]. The output phylogenetic tree with the domain structures of each member was finally edited using the ITOL tool (http://itol.embl.de). 

### 2.3. Expression Pattern in Different Tissues

The expression pattern of co-orthologous SWEETs pairs was analyzed at the transcript level by retrieving the Illumina RNA-Seq data that have been reported previously (Appendix A) [52]. The values of FPKM from different tissues including two batches of 7-week-old root (root1, root2) and leaf (leaf1, leaf2); the same stages for the stems and flowers, the silique 15 days after pollination, and a callus generated by tissue culture; were log2-normalized and displayed using the Multi-Experiment Viewer (MeV 4.8) program (http://mev.tm4.org). After that, Pearson correlation coefficient analyses were performed by R program. 

### 2.4. Plant Materials, Growth Conditions, and Stress Treatment 

Chinese cabbage seeds (cultivar Chiifu-401-42) were surface-sterilized with 70% ethanol for 1 min, vortexed in a 10% bleach solution 1500 rpm for 15 min, and then thoroughly washed with sterilized reverse-osmosis water 3–5 times before soaking overnight for two days in dark conditions at 24 °C. The germinated seeds were grown in half-strength Hoagland’s solution (pH = 6.0) in plastic containers and kept in the phytotron at the Nanjing Agricultural University at 70% relative humidity and 100 μmol m^−2^s^−1^ light intensity (light at 24 °C for 16 h, and darkness at 18 °C for 8 h). Five-leaved seedlings were accustomed to stress treatments by transferring the plants to chambers under the same conditions mentioned above but at different temperatures for the cold (4 °C) and hot (38 °C) treatments and with the replenishment of exogenous substances for the salt (200 mM NaCl) treatment. Samples with three biological replicates were taken after treatment at 0 h (CK), 1 h, 2 h, 4 h, 6 h, 8 h, and 10 h and immediately dropped in liquid nitrogen, and then they were stored at −80 °C for RNA isolation. 

### 2.5. RNA Isolation and qRT-PCR 

The extractions of the total RNAs and the syntheses of the first-strand complementary DNA (cDNA) were performed separately using a Total RNA Kit and an RNA TIANScript cDNA Synthesis Kit (samples; Tiangen Biotech, Beijing) according to the manufacturer’s guidance. The qRT-PCR was performed in a volume of 20 μL (containing 10 μL of SuperMix, 0.4 μL of each primer, 1 μL of the template (×10 diluted cDNAs), and 7.2 μL of sterile distilled water). The condition set-up was as follows: 95 °C for 30 s, followed by 40 cycles at 95 °C for 5 s, 55 °C for 15 s, and 72 °C for 10 s in the ABI StepOne Plus real-time PCR system. The relative gene expression analysis was counted using the 2^−ΔΔCt^ methods by using ACTIN and UBQ as reference genes [53] with the three independent replicates. All the primer pairs were designed with the Primer program (v5.0) (http://www.broadinstitute.org/ftp/pub/software/Primer5.0/) and they are listed in Appendix A.

## 3. Results

### 3.1. The Evolutionary Footprint of SWEETs in Plants

A comparative analysis of the molecular function and structural features between semiSWEETs in prokaryotes and SWEETs in eukaryotes illustrated that SWEETs expanded their functional capacity by enriching the structural features in eukaryotes [4]. Furthermore, mounting evidence shows that the extension of the SWEET gene family allows for diversification in metabolic regulation and specification in gene spatio-temporal expression [52,54,55,56,57]. Here, with the aim of investigating the footprint of the SWEET gene family in more widely distributed plant species, 486 homologs of SWEETs were retrieved using a BLASTP search (*E*-value ≤ 1e−10 and identity ≥ 60%) and MtN3/PQ-loop domain identification from 27 sequenced plant species in a different taxonomic group (Appendix A). The retrieved result showed that the SWEETs’ gene set was extended from land plants, and its isoform numbers are species-specific, ranging from six to 52 (Figure 1A). A statistical analysis on the transmembrane domain (TMs) suggested that many SWEET candidate genes have a certain sequence similarity by homologous search, but fewer than seven TMs also extended from land plants and tended to have more complex structural features from flowering plants. Since functionally verified structural features of SWEET proteins with two triple-helical bundles (THBs) are essential for functional sugar-translocating pores in eukaryotic organisms [4,58], the candidates (410 sequences) with two MtN3 domains were selected to further construct a phylogenetic relationship. Those sequences were phylogenetically divided into four groups (Appendix A). An orthologous analysis according to the phylogenetic relationship found that plant SWEETs may originate from Clade II, and they may have expanded from the vascular plant with the evolution of the vascular system, since more groups (Clade III and Clade IV) appeared from *Selaginella moellendorffii*. Clade I appeared soon afterward, from the flowering plant *Amborella trichopoda* (Figure 1B). Additionally, although SWEET genes set and functional differentiation expanded after land plants, they were seemingly barely related to the cyclic rounds of the genome duplication event (WGD or WGT) in the plant kingdom [59].

### 3.2. SWEETs Homologs in the B. rapa Genome 

The genome release of *B. rapa* confirmed that, other than sharing the three paleopolyploidy events (γ, β, and α) with the genome of *A. thaliana, B. rapa* suffered an additional WGT event after the divergence from *A. thaliana* 13–17 million years ago [34,43,60,61]. The diploidization process to follow involved genome reshuffling and gene losses, which have different effects on the retention of the gene family set. To analyze the duplication and retention pattern of SWEETs in *B. rapa*, the orthologs or co-orthologs of SWEETs in *A. thaliana* were investigated. Here, a total of 33 BrSWEET genes were first identified in the *B. rapa* genome according to their syntenic relationship and a BLASTp search with the best-matched hits for *A. thaliana* homologs (Appendix A) [5]. Each member was named on the basis of their orthologous relationship with *A. thaliana* and with co-orthologs in different sub-genomes by adding an LF, MF1, or MF2 suffix (Figure 2A). When mapping the SWEETs of *A. thaliana* and *B. rapa* onto 24 GBs of ancestral crucifer karyotype, all of them are mapped on 14 of the 24 GBs and they shared the selfsame GBs for orthologous genes distributed over eight ancestral crucifer karyotypes and expanding copies of AK1/3/5/6/7/8 in *B. rapa* (Figure 2B). By comparing them to the neighboring syntenic genes (10 genes on either side) in the flanking region, a higher proportion of BrSWEETs’ genes were retained for those homologs with two or three copies (Figure 2C), and more members were kept in the sub-genomes of MF1 and MF2 (Figure 2D).

After that, with the aim of understanding the species-specific amplification of SWEET genes in *B. rapa* better, basic information, including the phylogenetic relationships; copy number variations of orthologs; duplication event types; and their corresponding amino acid sequence alignment, TMs, conservative motifs, domains, and the exon-intron structures of homologs between *A. thaliana* and *B. rapa* were investigated. From an overall perspective, the result of a multiple sequence alignment of amino acid sequences illustrated that TM4 is less conservative for all the sequences (Figure 3B). All the BrSWEETs proteins were strictly conserved for the key functional sites, which validated that they were essential for structural formation, with sites such as Tyr57, Trp176, Val188, and four conserved prolines (P23, P43, P145, and P162) in AtSWEET1 [4], excepting isoforms with obvious fragment deletion (Appendix A). An analysis of the conservative motif and domain suggested that all the members of Clade III have longer protein sequences, which embed an extra motif 7 near the position of TM4, an extra motif 8 or motif 10 near C-terminal region, and its composed domains are closer to MtN3 than the PQ loop according to the sequence similarity retrieval prediction. 

A further analysis on the phylogenetic relationship, copy number variation, and duplication event type for SWEETs in *B. rapa* suggested that the case can be divided into three types as follows: (1) Removed, as with *AtSWEET6*: the only isoform with six TMs and without introns in *A. thaliana* loses its orthologs from the *B. rapa* genome; (2) single, such as tandemly duplicated gene pairs of *AtSWEET10*/*13*, *AtSWEET8,* and *AtSWEET9,* all of which left only one orthologous copy in the *B. rapa* genome (*BrSWEET8-LF, BrSWEET9-MF2,* and *BrSWEET10-MF1*), but kept their uniform structural features of seven TMs, motif distribution, two MtN3/PQ-loop domains, and exons/introns with orthologs in *A. thaliana*; (3) expanded, the remaining members are expanded with the *Brassica*-specific WGT event by two (*BrSWEET*1/2/4/5/7/12/*16/17*) or three (*BrSWEET3/11/14/15*) copies except for an extra tandem duplication event for *BrSWEET2-LFa/LFb* and *BrSWEET5-MF1a/MF1b*, respectively (Figure 3). 

For the expanded members, the amino acid sequence features and gene structure suggested that those co-orthologs had different fragments missing, although they still remained in the *B. rapa* genome. For the members with two copies that were duplicated by WGT, except for the co-orthologs (*BrSWEET4-LF* and *BrSWEET4-MF1*) of *AtSWEET4*, *BrSWEET1-LF*, *BrSWEET12-MF2*, and *BrSWEET16-MF2* that is missing motif 9. *BrSWEET7-MF1* is missing motif 2/3, and *BrSWEET17-MF2* is missing motif 3, when compared to their corresponding orthologs in *A. thaliana*. For tandemly duplicated co-orthologs of *BrSWEET2-LFa* and *BrSWEET2-LFb*, both have an obviously truncated sequence in the N and/or C-terminus of the flanking region, accompanying the missing conservative motifs, domain, and intron-exon structures at the corresponding region. For another pair of tandem duplicates, *BrSWEET5-MF1a* and *BrSWEET5-MF1b*, together with *BrSWEET5-MF2*, there was perfect structural consistency with the *AtSWEET5* ortholog for the motifs, domain, and intron-exon structure. For the members with three copies duplicated by WGT, who were co-orthologs of *AtSWEET3* and *AtSWEET15*, only one remained with structural consistency. The orthologs of *AtSWEET11* and *AtSWEET14* maintained three and two perfect structurally consistent regions, respectively, with the exception of *BrSWEET14-MF1*, which deleted the fragment on the N-terminus of the flanking region. Summarily, SWEETs did expand their family sets (194.12%) in the *B. rapa* genome, through integrating the WGT and tandem duplication event according to the identification of homologous genes, but they only duplicated co-orthologs of BrSWEET4, BrSWEET14, BrSWEET5, and BrSWEET11, which were strictly conserved with their orthologs in *A. thaliana*. 

### 3.3. In-Silico Assessment of Transcript Impact within Co-Orthologous BrSWEETs 

Since the BrSWEETs in *B. rapa* have rarely been studied, the transcript level of log2 transformed fragments per kilobase of the exon model per million mapped reads (FPKM) from different tissues including root, stem, leaf, flower, and silique [52], was employed to assess the transcript impact within amplified co-orthologous genes (Figure 4) with the aim of determining the expression pattern change for the retained SWEETs after amplification by a *Brassica*-specific WGT event or tandem duplication. The transcriptional abundance within co-orthologs suggested that most of the WGT-amplified co-orthologous BrSWEETs were prone to maintaining a very similar expression pattern, and they rarely showed obvious tissue-specific expression patterns. For instance, co-orthologs with three copies of *BrSWEET14* are comparatively highly expressed in the tissues of the root and flower, and the flower and silique, while *BrSWEET15* is primarily expressed in the flower and silique. Co-orthologs with two copies of BrSWEET4 are both highly expressed in flowers, and BrSWEET1, BrSWEET12, and BrSWEET17 are expressed in multiple tested tissues with differential transcript abundance. The convergence in the expression profile within those WGT-amplified co-orthologous SWEET genes suggested that some of them may coordinate their expression towards an abundant final product. Interestingly, for two pairs of tandemly duplicated SWEETs that were structurally characterized, *BrSWEET2-LFa* and *BrSWEET2-LFb* had the obvious essential domain/motif deficiency mentioned above, but their transcripts were detectable in some tissues. *BrSWEET5-MF1a* and *BrSWEET5-MF1b* displayed their complete structural features but their transcripts were almost undetectable in all the investigated tissues. There was a phenomenon in which there was an incompletely conserved domain/motif but a detectable transcript for co-orthologs, not only for tandemly duplicated members but also WGT-amplified members, which suggested that the strictly conserved domain was probably not directly related to the transcript. In addition, only *BrSWEET3-MF2* was found to have root-specific expression, and *BrSWEET5-MF2* is exclusively expressed in the silique, although three co-orthologous copies were retained and their structural conservation was characterized differently. In addition, the PCC analysis for the detectable transcript members suggested that WGT-amplified co-orthologous SWEETs are transcriptionally correlated with others but not tandem-duplicated members. 

### 3.4. Expanded Co-Orthologous Gene Expression in Response to Abiotic Stress 

Generally, the duplicate genes retained after the fractionation process are supposed to produce advantageous or functional profitably [54], and they have particularly been hypothesized to act as standby members in the genome for stress survival by somehow responding to extreme environments. Sugar transporter SWEETs usually modulate the stress response osmotically and/or via sugar metabolism signaling by interfering with sugar allocation [55,62]. Therefore, to determine whether abiotic stress affects the WGT-amplified paralogous SWEETs genes in *B. rapa*, salt, cold, and heat stress experiments were performed on the WGT-duplicated co-orthologs of BrSWEET1, BrSWEET11, BrSWEET12, and BrSWEET17—which are highly expressed in leaf tissues, as indicated by the in-silico investigation above (Figure 5). The expression patterns of two BrSWEET1 co-orthologs suggested that *BrSWEET1-LF* and *BrSWEET1-MF1* are both significantly up-regulated after 2 h of salt and cold treatment, but only *BrSWEET1-MF1* was dominantly expressed in response to heat stress at the 2 h and 10 h time points, indicating that they are both transcriptionally detectable but have different expression responses to various stresses. For three co-orthologs of BrSWEET11, *BrSWEET11-LF* slightly responded to salt and cold treatment, and it was barely expressed in the heat treatment; *BrSWEET11-MF1* was strongly induced under all stress treatments, but *BrSWEET11-MF2* was only dominantly expressed under heat treatment after 8 h. In addition, two co-orthologs of BrSWEET12 are both induced under different stresses. Their expression pattern suggested that BrSWEET11-LF and the two co-orthologues of BrSWEET12 probably have a broader response to stress environments compared to *BrSWEET11-MF2*. Additionally, *BrSWEET17-MF1* was dominantly expressed in response to different stress treatments, but *BrSWEET17-MF2* were barely detected. 

## 4. Discussion 

Higher plants have updated their way of transporting the photosynthetic assimilates, not only through open plasma membranes (PMs), which are connected by the plasmodesmata bridge, but also through enclosed membranes harboring sugar transporters to balance sugar export and import relying on demand and supply [1,56,57,63]. The identification of SWEET proteins not only fills the missing step for the apoplastic pathway of the phloem loading and unloading process but also clarifies the confusion about the sugar efflux-involved sink organ’s development and nutrition communication between plants and pathogens. Therefore, both its molecular function and regulatory mechanism have been explored for biomass improvement and environmental adaption, which depend on allocation deficiency. 

In the current study, an orthologous analysis of SWEETs from different plant groups according to the phylogenetic relationship suggested that higher plant SWEETs probably originated from Clade II and expanded from vascular plants with the evolution of the vascular system, since more groups (Clade III and Clade IV) appeared from *S. moellendorffii* and Clade I appeared soon afterward from flowering plants in *A. trichopoda* (Figure 1B). Previous publications have documented that Clade III transports sucrose, and Clade I, II, and IV predominantly transport hexose [6,18,32]. Furthermore, PM-localized SWEETs with sucrose transport capacity are hypothesized to move sucrose out of leaves involving the long-distance transport of assimilation product by way of apoplastic phloem loading in the plant vascular system. Therefore, the expansion of SWEETs’ functions in vascular plants probably extended their lifestyle by updating the distribution passage of assimilation products. 

After that, 33 BrSWEET genes were identified from the *B. rapa* genome, and co-orthologs of SWEETs in *A. thaliana* were labeled by adding the sub-genome of the LF, MF1, or MF2 as a suffix (Figure 2A). Compared to the 34 SWEETs identified by Miao L et al. [64], BrSWEET3-MF1 (gene ID: Bra029090) is not thought to belong to the BrSWEETs family, because of the absence of any of MTs or MtN3/PQ-loop and because it barely qualified for phylogenetic analysis. The distribution of SWEETs in *A. thaliana* and *B. rapa* on 24 GBs of ancestral crucifer karyotypes indicated that all of them are mapped on 14 of 24 GBs and they shared the same GBs for co-orthologous SWEETs (Figure 2A). The expanded copies of BrSWEETs occurred at AK1/3/5/6/7/8 (Figure 2B), and a higher proportion of BrSWEETs genes were retained for homologs with two or three copies compared to the neighboring syntenic genes in the flanking region (Figure 2C). Furthermore, more members remained in the sub-genomes of MF1 and MF2 (Figure 2D). From an overall perspective of the identified SWEETs in *B. rapa*, TM4 is less conserved and all the members showed strict conservation of the key functional sites, except for isoforms with obvious fragment deletion. An analysis of the conserved motif and domain suggested that all the members in Clade III have longer protein sequences, which embed an extra motif 7 near the TM4 position; an extra motif 8 or motif 10 near the C-terminal region; and their composed domains are closer to MtN3 rather than the PQ-loop, according to the sequence similarity retrieval prediction. Further detailed analysis combining the copy number variation, duplication event type and sequence features suggested that *AtSWEET6* lost its orthologs in the *B. rapa* genome; tandemly duplicated *AtSWEET10* and *AtSWEET13*, leaving only one orthologous copy in the *B. rapa* genome; and most of the remaining amplified SWEETs in the *B. rapa* genome occurred by WGT or tandem duplication events, which left only one isoform with strictly conserved orthologs in *A. thaliana,* except for the co-orthologs of BrSWEET4, BrSWEET5, BrSWEET11, and BrSWEET14. Subsequently, the transcriptional level of BrSWEETs in different tissues illustrated that there was an incompletely conserved domain/motif but also a detectable transcript for co-orthologs; not only for tandemly duplicated members but also WGT-amplified members, indicating that the strictly conserved domain was probably not directly related to the transcript. Further analysis of the expression levels under salt, cold, and heat stress conditions for WGT-duplicated co-orthologs of BrSWEET1, BrSWEET11, BrSWEET12, and BrSWEET17, which are highly expressed in leaf tissues according to in-silico investigation, showed that *BrSWEET11-LF* and the two co-orthologs of BrSWEET12 probably provide broad responses to stressful environments compared to the others, which have specific responses to different stresses. 

In summary, from the evolutionary footprint of the SWEETs, although the SWEETs gene set and functional differentiation expanded after land plants occurred, it was seemingly barely related to the cyclic rounds of genome duplication events (WGD or WGT) in the plant kingdoms. However, for the recent WGT event in the *B. rapa* genome, more WGT-duplicated SWEETs were retained for amplifying the SWEETs family set, and their transcripts are detectable. The convergence or divergence in the expression of co-orthologous SWEETs’ responses to biotic stress implies functional diversity. This study provided a detailed investigation of the SWEETs in *B. rapa*, and it also raises a number of questions. For example, the relationship is between TM4 and motif 7 and whether motif 7 is necessary for sucrose transport, since it was specifically present in Clade III. This early work on SWEETs in *B. rapa* will fundamentally assist in further molecular function characterizations and will help during explorations of genetic innovation.

## Figures and Tables

**Figure 1 genes-10-00722-f001:**
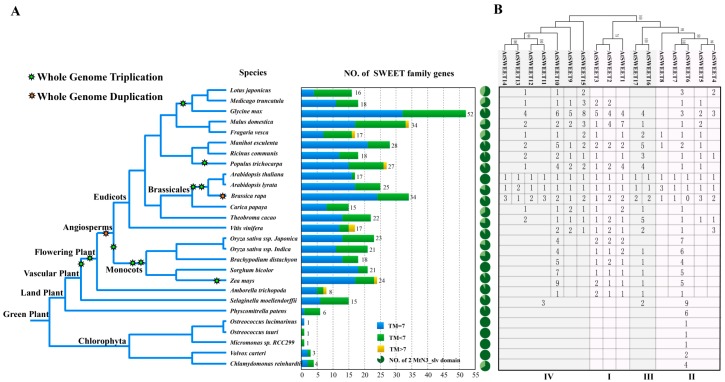
Molecular footprint and phylogenetical classification of SWEETs in the plant kingdom. (**A**) Identification and distribution of homologous SWEETs in different plants and (**B**) functional classification based on maximum likelihood (ML) phylogenetic analysis by Appendix A.

**Figure 2 genes-10-00722-f002:**
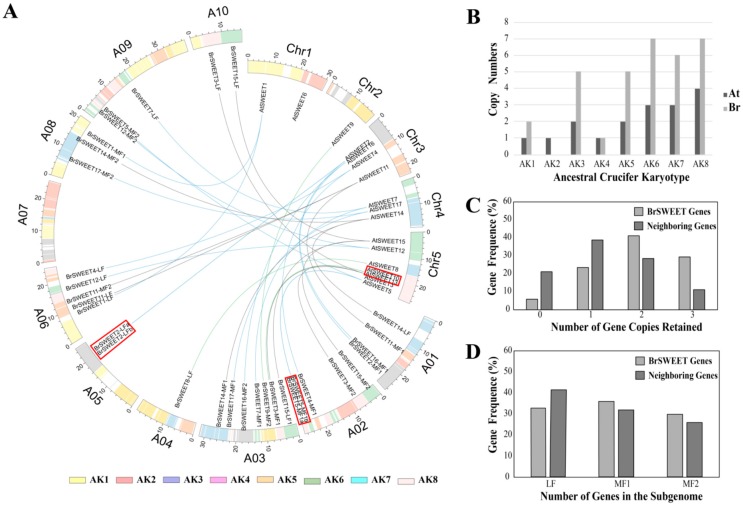
Basic information of the SWEET genes in the *Arabidopsis thaliana* and *Brassica rapa* genome. (**A**) Assignment of the SWEET genes to the chromosomes and 24 gene blocks (GBs) of *thaliana* and *B. rapa*; (**B**) SWEETs numbers on the inferred ancestral crucifer karyotype; (**C**) gene frequencies of retained homologous copies in the syntenic region; and (**D**) three sub-genomes with SWEETs in *B. rapa*. The red box labeled gene pairs in (**A**) expanded by tandem duplication and the fractionation of SWEETs homologs in different sub-genome are listed in Appendix A. Different colored lines indicate the orthologous SWEETs (green for single copy, blue for two copies, and black for three copies) between *A. thaliana* and *B. rapa*.

**Figure 3 genes-10-00722-f003:**
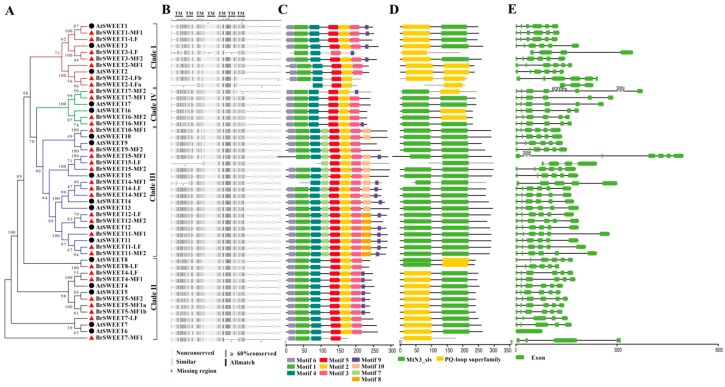
Sequence feature analysis of SWEETs in *A. thaliana* and *B. rapa*. (**A**) NJ-phylogenetic relationship, (**B**) sequence alignment of amino acid, (**C**) conserved motif, (**D**) conserved domain, and (**E**) exon-intron structure. The seven transmembrane™ domains were assigned a sequence alignment based on the AtSWEET1 structure indicated by Tao et al. [4].

**Figure 4 genes-10-00722-f004:**
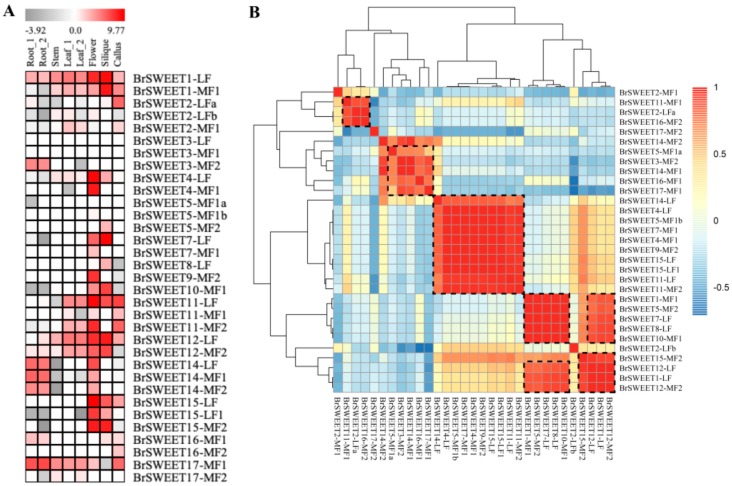
Heat map (**A**) and Pearson correlation coefficient (PCC) analysis (**B**) of the SWEET gene transcriptional levels in different tissues of *Brassica rapa*. The dotted black box indicates the members with transcriptional correlation. Blocks with colors (**A**) indicate low (dark) to high (red) normalized transcript accumulation and blocks, with colors in (**B**) indicating a negative PCC correlation (−1) to positive PCC correlation (1).

**Figure 5 genes-10-00722-f005:**
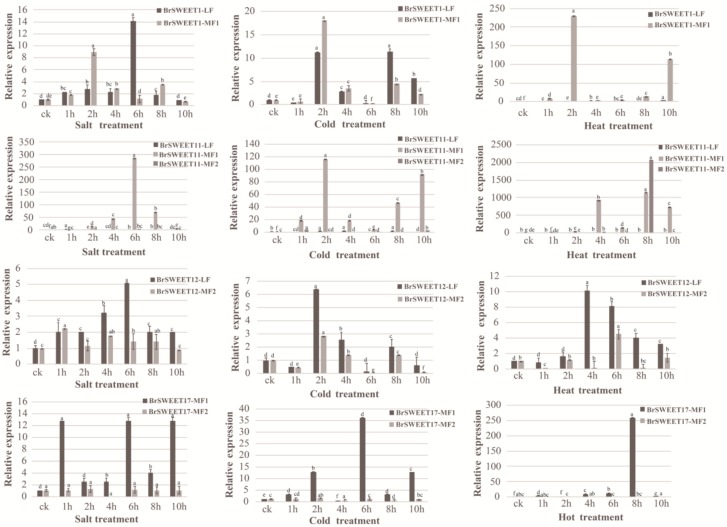
Expression patterns of co-orthologs of SWEETs under salt, cold, and heat stress treatments in *B. rapa* by RT-qPCR. Error bars represent ± SE (*n* = 3). The one-way analysis of variance was calculated by Duncan’s new multiple range test, *n* = 3; different letters above the bars indicate significant differences (*p* < 0.05).

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
