# Peer review of "The Expanded SWEET Gene Family Following Whole Genome Triplication in Brassica rapa"

_genes, 2019, doi:10.3390/genes10090722_

Round 1

Reviewer 1 Report

Given my last review of minor revisions, I believe the editors can
decide if they are willing to publish the current version of the
manuscript.

Author Response

Response: Thanks for the positive feedback. The abstract part has been rewritten, and the English spell and style have carefully checked in the update version of manuscript.

Reviewer 2 Report

The authors have addressed my prior concerns.

Author Response

Response: Thanks for kindly support. Minor spell was carefully checked in the revised version.

This manuscript is a resubmission of an earlier submission. The following is a list of the peer review reports and author responses from that submission.

Round 1

Reviewer 1 Report

Wei et al., reported the identification of SWEET gene family in Brassica rapa, and trying to discuss their behavior during whole genome triplication. Their phylogenetic analyses generally worked out, but more details need to be provided. They also attempted to assay the expression of SWEET genes under stress conditions by qPCR. This part of study is lack of biological replicate and statistical test, which needs to be modified.

1.There are many studies regarding the identification of SWEET genes in different organisms. The authors may consider compare their results in (e.g. rice, Arabidopsis) with genes identified in this study, especially the article about the identification of SWEET in B. rapa by Li et al., in FIPS.

2.The authors need to adjust the size and resolution of all figures. There are too tiny to be recognized.

3. Figure 2:

a.It would be better if segmental duplications are labeled in Figure 2.

b.It is unclear what the blue color and arrow refer to.

c.It is unclear how the gain/loss/synteny of neighbouring genes are defined.

d.From Figure 2D, it seems that LF and MF1 have more genes than MF2, different from the author’s description (L138).

4.Figure 3: the authors may consider to explain why MtN3 are counted as a different domain as PQ-loop (also in L157). From the introduction, it seems that they are counted as the same domain.

5.Duplication event type:

This paragraph needs to be better demonstrated. An extra figure to explain the mode of operation would be good. In addition, there are 34 genes in Table S2, but only 33 were reported in the article (L128). Why the second BrSWEET10 is not labeled in Figure 2? Furthermore, the ‘solo’ type is confusing. AtSWEET10/13 are tandemly duplicated and 13 is gone in B.rapa while 10 has two orthologs. It is unclear how the solo is defined. And is more confusing consider SWEET 8/9 is not tandem duplicated.

6. L172-188 does not have any figures to support.

7. Heatmap:

a. The authors need evidence to support their claim about ‘a quite similar expression pattern’. Maybe test for correlation. In addition, there is no criteria for defining ‘multi-tissues with different transcript abundance’ (L202). It could not be based on eye-balling a heatmap. 

b. The authors may consider performing analyses on comparing expression levels of different subgenomes. For example, in maize, subgenome 1 is dominant in expression compared to subgenome 2.

8. For qPCR, biological replicates and statistical tests should be included. And it would be easier if the bar graph shows the ratios of treatment (1-10h) to control(CK).

Author Response

Notes to Reviewer #1

>>>Comments and Suggestions for Authors

Thank you for this valuable feedback, we are extremely grateful to you for pointing out the specific detail problems for our paper. We would like to adjust the manuscript accordingly for an improved version.

>>>Wei et al., reported the identification of SWEET gene family in Brassica rapa, and trying to discuss their behavior during whole genome triplication. Their phylogenetic analyses generally worked out, but more details need to be provided. They also attempted to assay the expression of SWEET genes under stress conditions by qPCR. This part of study is lack of biological replicate and statistical test, which needs to be modified.

Response: Thanks for the reviewer’s comments on our manuscript. We carefully checked the details and revised for phylogenetic analysis part to make it clearer in the revised manuscript. And we do have the biological replicate and statistical test, sorry for the confusion and we have corrected the description on the method and result involving in the qPCR part.  

>>>1.There are many studies regarding the identification of SWEET genes in different organisms. The authors may consider compare their results in (e.g. rice, Arabidopsis) with genes identified in this study, especially the article about the identification of SWEET in B. rapa by Li et al., in FIPS.

Response: Thanks for your suggestion. Firstly, I am sorry for the original manuscript which missed Figure 1 and not sure whether you got an updated version. Figure 1 in this manuscript, includes the summary of identification and phylogenetical classification for SWEETs in different plant species (detailed phylogenetic relationship and information attached in the supplementary files). There had a rough comparison with other plant and a detailed comparison with A. thalian since an extra WGT event for B. rapa comparing to A. thaliana. Besides, for the identification of SWEET in B. rapa by Li et al., in FIPS, we did read it when it just published and honestly, we finished this part earlier and we even had a very comprehensive comparison for the identification from three versions of B. rapa genome (v1.5, v2.0 and v3.0). We admit there no scientific innovation for identification of SWEETs in B. rapa, but still think that a basic and comprehensive understanding is quilt necessary for the story in this manuscript and our whole project.

>>>2.The authors need to adjust the size and resolution of all figures. There are too tiny to be recognized.

Response: Thanks for the advice about figures, the words size, and picture resolution have been adjusted in the revised manuscript for clearer recognition.

>>>3. Figure 2:

a. It would be better if segmental duplications are labeled in Figure 2.

b. It is unclear what the blue color and arrow refer to.

c. It is unclear how the gain/loss/synteny of neighboring genes are defined.

d. From Figure 2D, it seems that LF and MF1 have more genes than MF2, different from the author’s description (L138).

Response: Thanks for the advice, we have corrected the section accordingly in the result and method section of Figure 2 in the revised manuscript.

4.Figure 3: the authors may consider to explain why MtN3 are counted as a different domain as PQ-loop (also in L157). From the introduction, it seems that they are counted as the same domain.

Response: Thanks, it did report that PQ-loop and MtN3 are the similar domain (generally roughly termed together as PQ-loop/MtN3/saliva/SWEET) in some references since it is difficult to clearly distinguish the PQ-loop and MtN3 domain according to sequence identification (such as Yuan M et al.' report in the rice which published at Mol Plant, 2013, 6: 665–674). However, from the previous reports (like semiSWEETs in prokaryotes contains more PQ-loop than MtN3) and our broader SWEETs analysis in the plant kingdom (result not shown in present manuscript), MtN3 domain tends to be evolved from PQ-loop and somehow mainly response for sucrose metabolism in clade III but there more pieces of evidences are needed. Therefore, in the present manuscript, we prefer to keep the result which predicted by the same algorithm mechanism, but we would like to adjust the description to make it clearer in the revised manuscript.     

>>>5.Duplication event type:

This paragraph needs to be better demonstrated. An extra figure to explain the mode of operation would be good. In addition, there are 34 genes in Table S2, but only 33 were reported in the article (L128). Why the second BrSWEET10 is not labeled in Figure 2? Furthermore, the ‘solo’ type is confusing. AtSWEET10/13 are tandemly duplicated and 13 is gone in B. rapa while 10 has two orthologs. It is unclear how the solo is defined. And is more confusing consider SWEET 8/9 is not tandem duplicated.

Response: Thanks for the comment.

For the question "three are 34 genes in Table S2, but only 33 were reported in the article (L128). We did identify 34 genes but one was filtered out since lack of any MtN3 or PQ-loop domain (Line)

For the question "Why the second BrSWEET10 is not labeled in Figure 2? Furthermore, the ‘solo’ type is confusing. AtSWEET10/13 are tandemly duplicated and 13 is gone in B. rapa while 10 has two orthologs". Sorry for the confusion about this part, AtSWEET10 and AtSWEET 13 are tandemly duplicated in A. thalian but just one of them kept in the B. rapa genome (named as BrSWEET10-MF1 in manuscript) and the sentence has revised for this part in the revised manuscript.

    For the question "It is unclear how the solo is defined. And is more confusing consider SWEET 8/9 is not tandem duplicated". Sorry for the confusing description. AtSWEET8 and AtSWEET9 are not tandem duplicated, the sentence has revised in the manuscript.

>>>6. L172-188 does not have any figures to support.

Response: Thanks, the information has been added accordingly.

>>>7. Heatmap:

>>>a. The authors need evidence to support their claim about ‘a quite similar expression pattern’. Maybe test for correlation. In addition, there is no criteria for defining ‘multi-tissues with different transcript abundance’ (L202). It could not be based on eye-balling a heatmap.

Response: Thanks, the correlation analysis has been added in Figure 4 and the sentences had been corrected accordingly.

>>>b. The authors may consider performing analyses on comparing expression levels of different subgenomes. For example, in maize, subgenome 1 is dominant in expression compared to subgenome 2.

Response: Thanks for the suggestion to compare the expression levels of different subgenomes. We are very interested the expression pattern for the WGD/WGT species in different subgenomes since we had a broader bioinformatics analysis in the plant kingdom for the SWEETs evolution in the phloem loading process and a comprehensive investigate for SWEET gene family amplification in crucifers' group. However, until now we haven't got a very clear point for drifting a manuscript since the data analysis part not finish yet. Can you please follow our lab's further report if you are interested in it as well?

>>>8. For qPCR, biological replicates and statistical tests should be included. And it would be easier if the bar graph shows the ratios of treatment (1-10h) to control (CK).

Response: Thanks, we do have the biological replicates and statistical test, sorry for the confusion and we have corrected the description on the method and result involving in the qPCR part.  

Reviewer 2 Report

The paper entitled "The Expanded SWEET Gene Family Following Whole Genome Triplication in Brassica Rapa" by Wei, et al. surveys the phylogenetic relationship of SWEETs across plants with a specific focus on the evolution of this gene family in Brassica rapa. In addition to careful phylogenetic analysis of this gene family, the authors survey expression profiles of Brassica rapa SWEETs both across tissues and in various conditions to provide a first look at the potential role of each family member

Overall, the types and depth of analyses performed in this manuscript are similar to others recently published in this journal. I have no major concerns with the analytical approaches taken by the authors as outlined in the Materials and Methods. The clarity of the manuscript needs to be substantially improved, however. In its current state, the key points of the manuscript are difficult to discern as one out of every three or four sentences requires revision. In particular, the sections in the Introduction and Results (2.2) discussing WGD in B. rapa were difficult to follow. I don't feel that I can provide a critical review of the manuscript in its current form. 

Author Response

Notes to Reviewer #2

>>>Comments and Suggestions for Authors

>>>The paper entitled "The Expanded SWEET Gene Family Following Whole Genome Triplication in Brassica rapa" by Wei, et al. surveys the phylogenetic relationship of SWEETs across plants with a specific focus on the evolution of this gene family in Brassica rapa. In addition to careful phylogenetic analysis of this gene family, the authors survey expression profiles of Brassica rapa SWEETs both across tissues and in various conditions to provide a first look at the potential role of each family member

Overall, the types and depth of analyses performed in this manuscript are similar to others recently published in this journal. I have no major concerns with the analytical approaches taken by the authors as outlined in the Materials and Methods. The clarity of the manuscript needs to be substantially improved, however. In its current state, the key points of the manuscript are difficult to discern as one out of every three or four sentences requires revision. In particular, the sections in the Introduction and Results (2.2) discussing WGD in B. rapa were difficult to follow. I don't feel that I can provide a critical review of the manuscript in its current form.

Response: Thanks for carefully and patiently reviewing our manuscript.

First of all, we are sorry for giving a poor-quality manuscript with the English language and style mistakes or confuse. We have carefully checked and revised the paper with the help of two native English speakers, hope that the revised version is more readable.

Next, for the SWEET in B. rapa which have been published, we admit there no scientific innovation for identification of SWEETs in B. rapa, but still think that a basic and comprehensive understand is quite necessary for the story in this manuscript and our whole project. We even had a very comprehensive analysis to compare the identification result from three versions of B. rapa genome (v1.5, v2.0, and v3.0) since our following experimental result for molecular function has been confused us for some members. 

Last, we have revised the sections in the Introduction and Results (2.2) in the revised version. 

Round 2

Reviewer 1 Report

The authors have resolved some of my concerns. However, some points still awaits to be corrected.

1. The quality of figure 1,2,3 and 5 still needs to be improved. I could barely recognize some of the contents.

2. The authors did not respond to the segmental duplication in Figure 2 and how they define gain/loss/synteny of neighboring genes. Nor did they respond to why MF1 and MF2 have more genes than LF.

3. For heatmap, although the authors perform a correlation analysis in Figure 4, it did not prove ‘co-orthologous genes have similar expression’. Some co-orthologous are not clustered together (e.g. sweet 5). They also need to describe the numbers in the figure legend. Furthermore, I think it is a better idea to calculate the correlation of expression levels between/among co-orthologous genes to support their hypothesis.

4. The authors did not change the expression level to ratios as I suggested, nor did they respond to why they did not change. Meanwhile, I did not find where are the statistical tests as they claimed. Finally, the error bars should be based on bio reps not tech reps.

Reviewer 2 Report

Upon re-review of the of the manuscript entitled "The Expanded SWEET Gene Family Following Whole Genome Triplication in Brassica rapa" by Wei, et al. I find that the authors did attempt to improve the clarity of their writing. I don't believe the language quality should prohibit the publication of this work but the manuscript would still benefit from professional editing. As I said before, I find no major problems with their methodology and the scope of their work is similar to others published in this journal.